# Millisecond cryo-trapping by the spitrobot crystal plunger simplifies time-resolved crystallography

Pedram Mehrabi [1,2,5] ✉, Sihyun Sung [3], David von Stetten [3],
Andreas Prester [4], Caitlin E. Hatton [1], Stephan Kleine-Döpke [1],
Alexander Berkes [1], Gargi Gore [1], Jan-Philipp Leimkohl [2], Hendrik Schikora[2],
Martin Kollewe[2], Holger Rohde [4], Matthias Wilmanns [3,4],
Friedjof Tellkamp [2] ✉ & Eike C. Schulz [1,2,4,5] ✉

We introduce the *spitrobot*, a protein crystal plunger, enabling reaction quenching via cryo-trapping with a time-resolution in the millisecond range. Protein crystals are mounted on canonical micromeshes on an electropneumatic piston, where the crystals are kept in a humidity and temperature-controlled environment, then reactions are initiated via the liquid application method (LAMA) and plunging into liquid nitrogen is initiated after an electronically set delay time to cryo-trap intermediate states. High-magnification images are automatically recorded before and after droplet deposition, prior to plunging. The SPINE-standard sample holder is directly plunged into a storage puck, enabling compatibility with high-throughput infrastructure. Here we demonstrate binding of glucose and 2,3-butanediol in microcrystals of xylose isomerase, and of avibactam and ampicillin in microcrystals of the extended spectrum beta-lactamase CTX-M-14. We also trap reaction intermediates and conformational changes in macroscopic crystals of tryptophan synthase to demonstrate that the spitrobot enables insight into catalytic events.

Proteins are vital to all processes of life and during the last decade technological advances have enabled transformative changes in our ability to determine structural changes *in-operando*. While traditionally, trapping approaches were exploited to derive kinetic information from crystallographic experiments, a more recent development in this regard is time-resolved serial crystallography, which offers unprecedented structural insight into *out-of-equilibrium* conformations and reaction intermediates that cannot be provided by other methods[1,2]. Typically, a reaction is initiated in a protein crystal, which is subsequently exposed to an X-ray pulse after a defined delay time. This procedure is repeated for thousands of crystals, which eventually

yields a 3D structure of this time point. Several of such temporal snapshots can then be assembled into 'movies', providing even insight into the fastest processes of life[3].

However, uncovering these details requires insight from a variety of fields, which is typically beyond the scope of a non-specialist group. There is often an unfortunate gap between technological and scientific expertise that is required to carry out these experiments. How much more could be learned about the processes of life, if it was experimentally more accessible?

This dilemma becomes even more pronounced when taking into account that the majority of enzymes display only moderate turnover

[1]Institute for Nanostructure and Solid-State Physics, Universität Hamburg, Hamburg, Germany. [2]Max Planck Institute for the Structure and Dynamics of Matter, Hamburg, Germany. [3]European Molecular Biology Laboratory, Hamburg Unit, Hamburg, Germany. [4]University Medical Center Hamburg-Eppendorf (UKE), Hamburg, Germany. [5]These authors contributed equally: Pedram Mehrabi, Eike C. Schulz. ✉e-mail: pedram.mehrabi@uni-hamburg.de; friedjof.tellkamp@mpsd.mpg.de; ec.schulz@uke.de

kinetics of ~10 s⁻¹, making them accessible by synchrotron radiation experiments[4]. This means there is a large number of systems that could be probed by readily available facilities. Primarily aiming for these 'biologically relevant' time scales, we have recently developed the *hit-and-return* (HARE) and the liquid application method for time-resolved analyses (LAMA) for in-situ mixing[5,6]. The versatility of these methods encouraged us to further facilitate time-resolved experiments and make them more accessible to the large user base that has already access to standardized tools for high-throughput macro-molecular crystallography at synchrotron beamlines. Bearing in mind that most structure solutions by X-ray crystallography are carried out at cryogenic conditions, and can be done remotely, we sought for a solution that would bridge the gap between traditional and time-resolved methods using resources available to most structural biology labs.

Traditionally cryo-trapping has been exploited to quench enzymatic reactions within protein crystals and thereby obtain structural information about reaction intermediates[1]. However, these traditional approaches suffer from the limitations of manual substrate deposition and accurate, reproducible delay times, in particular with respect to fast time scales. To this end, we developed the *spitrobot* crystal plunger, which enables cryo-trapping experiments with versatile time-resolutions down to the millisecond range via the LAMA method (Fig. 1).

## Results

The *spitrobot* comprises several different, main hardware parts: (a) the plunger, (b) the humidity flow device (HFD), (c) the LAMA droplet injector, (d) the vitrification chamber, (f) the camera system, and (e) the control unit. All parameters are set via a control software.

Conceptually similar to cryo-EM vitrification devices[7], the *spitrobot* relies on crystals mounted onto SPINE-standard MicroMesh™ sample holders[8]. To trap reaction intermediates the micromeshes with protein crystals are mounted on an electropneumatic piston in a humidity and temperature-controlled environment. A sequence of electronic signals initiates the in situ mixing reaction by shooting a burst of picoliter-sized droplets onto the mesh-mounted crystals using our established LAMA technology[5]. After a defined delay time, the micromeshes are directly plunged into SPINE-standard pucks submerged in liquid nitrogen (Fig. 1). As a quality control, sample images are automatically acquired before and after droplet deposition. Adhering to the SPINE standard simplifies the integration into established high-throughput beamline workflows.

## The plunger

The main component of the plunger is an electropneumatic piston that drives the sample into the liquid nitrogen (Supplementary Fig. 1). It is mounted on a sturdy steel post on top of the vitrification chamber. The plunging velocity is regulated via the applied gas pressure. For typical use we relied on pressure levels from 3 to 6 bar, which enable piston motions on the order of 1.6 m s⁻¹, comparable to previously published solutions[9,10]. The piston is equipped with an electromagnetic SPINE-style sample holder, onto which the micro-meshes are manually mounted. After being submerged in liquid nitrogen the micro-meshes are automatically released into the SPINE-pucks, minimizing manual interaction after sample preparation. For reaction initiation the LAMA nozzle needs to be positioned within ca. 1 mm of the micro-mesh. To avoid accidental collisions and simplify sample mounting, the LAMA nozzle is retracted via rail-mounted translation stages. Once the sample is mounted, the LAMA nozzle is pushed back into place, and subsequently fine-aligned to the micro-mesh. Using the SPINE standard provides a number of advantages regarding the compatibility with established high-throughput beamline workflows. Adhering to this established standard will streamline crystal storage and shipment from

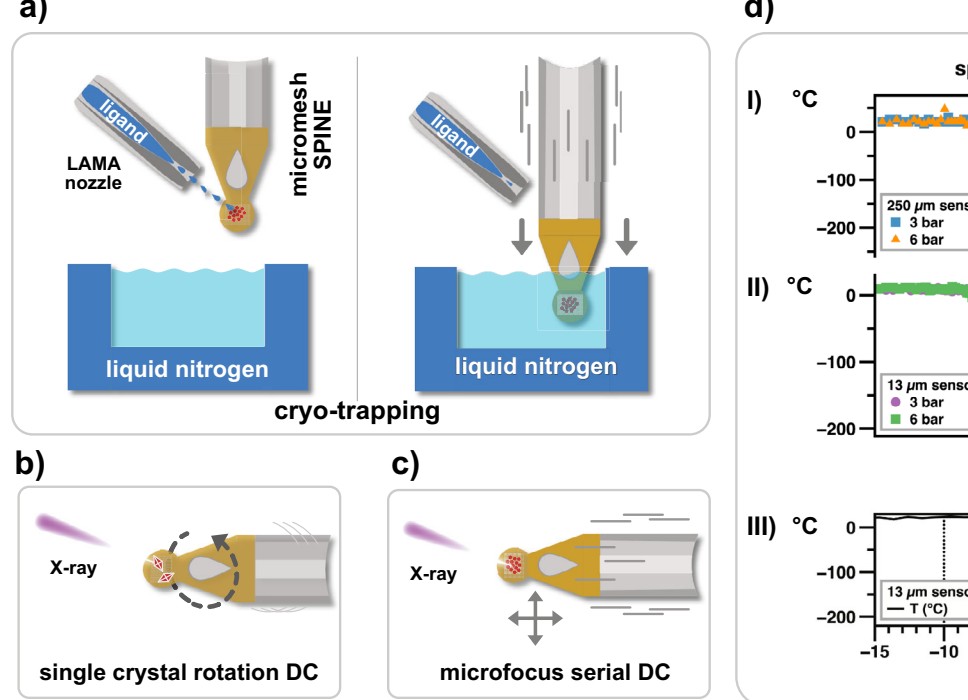

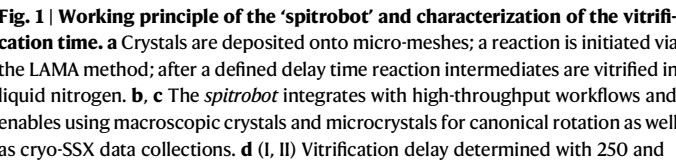

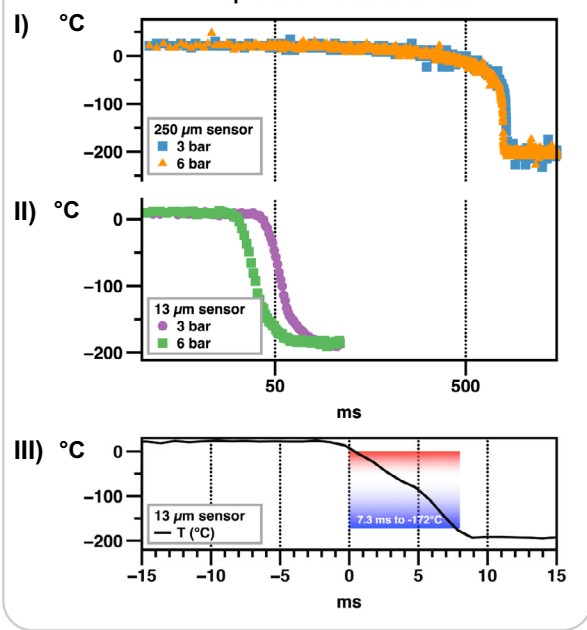

**Fig. 1 | Working principle of the 'spitrobot' and characterization of the vitrification time. a** Crystals are deposited onto micro-meshes; a reaction is initiated via the LAMA method; after a defined delay time reaction intermediates are vitrified in liquid nitrogen. **b, c** The *spitrobot* integrates with high-throughput workflows and enables using macroscopic crystals and microcrystals for canonical rotation as well as cryo-SSX data collections. **d** (I, II) Vitrification delay determined with 250 and 13 μm temperature sensors, respectively, matching the size range of typical samples; (III) an experimental characterization of the vitrification time demonstrates that the glass-transition temperature (160 K) of a typically-sized sample is reached within ~7.5 ms. The total minimal delay for microcrystalline samples is approximately 50 ms. Plunging time is affected by the air-pressure of the piston (indicated in bar).

(remote) MX-labs to synchrotron facilities, where sample exchange and automated data-collection procedures also rely on standardized samples, greatly increasing the turnover of samples and even enabling fully automated data-collection[11].

## Environmental control with the HFD

To maintain the crystals in an environment close to physiological conditions with controlled humidity and temperature we have developed a Humidity Flow Device (HFD), which provides a humid airflow at a defined temperature between 4 °C and 40 °C (Supplementary Fig. 2). Both the relative humidity and the temperature can be adjusted independently. The nozzle of the HFD has an inner diameter of about 13 mm and is placed about 1 cm from the SPINE sample. On the inside of the nozzle, a few cm before its end, is a combined sensor for humidity and temperature whose signals are transmitted to a microcontroller (Arduino nano), which calculates the control variables. The temperature is controlled by heating resistors; the humidity is controlled by a warm water bath, which is equipped with ultrasonic nebulizers. An external cooler can also be connected via a heat exchanger. The HFD provides temperatures between 4 °C and 40 °C at a humidity of up to 99%, with typical flow rates between 20 and 35 l/min. Humidity control also permits controlled crystal dehydration if required (Supplementary Information). To characterize the stability of the HFD we recorded step-functions of the relative humidity and the temperature, respectively, as a function of time. Humidity was increased at 5% increments and maintained at stable flow for several minutes. After an equilibration period, the relative humidity can be maintained within less than a percent (Supplementary Fig. 3), demonstrating its suitability to maintain a stable humidity environment for crystals and micro-crystals during sample preparation (Supplementary Fig. 4a). For the temperature step function, the temperature was recorded at 4 °C, 10 °C, 20 °C, 30 °C, and 40 °C for several minutes, respectively, keeping the relative humidity constantly at or above 95%.

## Reaction initiation

The LAMA droplet injector has been previously described in detail[5]. Briefly: via a piezo-actuator, 75 or 150 picoliter-sized droplets are shot from a 50 or 70 μm (inner diameter) glass capillary with a velocity of 2 m s$^{-1}$ onto the target mesh. The nozzle is brought close (1–2 mm) to the target via manual, rail-mounted translation stages that enable precise lateral and vertical alignment of the nozzles to adjust for differences in sample holder length and also to correct for bent meshes. Nozzle alignment is aided by two perpendicularly aligned cameras, which focus on the target mesh (see below). This way the nozzle distance, as well as its lateral and vertical alignment can be precisely controlled and adjusted to individual samples. Since the micromeshes provide a large sample area, a high-frequency (5 kHz) burst of picolitre droplets are added to the samples. The total volume of required liquid depends on the sample area that has to be covered, the protein and ligand concentrations, and the viscosity of the solutions. Between 100 and 500 droplets were applied for each sample used in this study corresponding to a volume between 15 and 75 nl, respectively.

## The vitrification chamber

The vitrification chamber is comprised of a regular foam dewar into which an aluminum mount for the SPINE standard puck is fixed. The mount permits a step-wise rotation of the puck between its 10 positions for sample vials for aligning each position to the vitrification point of the *spitrobot*. Rotation of the puck to the next sample position is done manually via a hexagonal bolt screw driver, each position snaps into place. This simplifies and accelerates sample handling and transfer as the process of vitrification deposits the sample directly into the puck, enabling further usage of high-throughput infrastructure. The dewar is closed via a transparent poly- methyl-methacrylate (PMMA) lid, with an opening for the piston. To reduce icing in the liquid nitrogen phase and improve vitrification rates the gaseous nitrogen layer that forms between the lid and the liquid phase is displaced by a stream of dry nitrogen gas as demonstrated previously[9]. To this end, the dry nitrogen is actively siphoned away via a connected pump. To further reduce ice formation on the lid, it is heated via a resistor array. Liquid nitrogen is replaced manually at regular intervals via the refill hole or the piston opening.

## Experimental characterization of the vitrification process

The vitrification time was characterized by two independent approaches, optically and electronically. For an optical characterization an LED flashing every 2.5 ms (400 Hz) was mounted to the tip of the piston. A long time-exposure synchronized to the plunging process captured the number of flashes during the piston motion. Since 9 flashes were recorded, this is equivalent to a piston motion time of 22.5 ms. However, this procedure only characterizes the piston motion, thus the switching delay of the air valve and the actual vitrification time were addressed in a separate experiment.

To obtain accurate vitrification times we used a thermocouple with similar dimensions to the crystal size. For comparison to previous studies, which mainly used larger thermocouples we thus recorded vitrification times using two temperature sensors of different size. Taking into account the *spitrobots* total processing time, including air-valve delay, plunge time, and vitrification, the larger RTD sensor (3.0 × 0.8 × 0.25 mm (IST, P1K0.308.7W.B.007 Farnell, Germany, −200 °C–600 °C) displays a total time of 800 ms, due to a large offset, which is presumably due to the Leidenfrost effect (insulating vapor layer between the sample and the cryogenic liquid), as the temperature drop-off from quenching is quite quick once the insulating vapor layer has dissipated. By contrast, the smaller thermocouple, which has approximately the same dimensions as the samples of interest (K-type thermocouple [KFT-13-200-200(Y)], ~13 μm diameter, ANBE SMT Co., Osaka, Japan) minimizes this offset and therefore displays a total processing time of ~50 ms and is thus almost negligible for the relevant time scales (Fig. 1d).

To determine the actual vitrification time, we also recorded the temperature decrease independent of the spitrobot. Here the glass-transition temperature (<−140 °C) is reached within 7.5 ms and a 90–10% analysis of the temperature drop resulted in a fall time of 7.5 ms. Thus, the total delay time of the *spitrobot* consists of the intrinsic delay of the device (air valve delay ~20 ms, piston motion ~25 ms), and the actual vitrification process ~7.5 ms corresponding to a cooling rate of $2.3 \times 10^4$ Ks$^{-1}$. These vitrification times are comparable to those reported previously for flash cooling devices operating with liquid nitrogen[9]. Based on these observations the dead-time of the *spitrobot* is on the order of ~45 ms and time-points with a minimal delay time of approximately 50 ms can be obtained.

Characterization of the processes demonstrates a vitrification time of ~7.5 ms, for samples on the order of 10 μm, and that physiological conditions can be maintained within the sample area and (Fig. 1, Supplementary information). The minimal delay time currently achievable with the *spitrobot* setup is ~50 ms. This constraint is the result of the mechanical piston motion (22.5 ms), air-valve delay (20 ms), and vitrification time (7.5 ms)—the deposition of the substrate solution occurs during the air-valve delay and is thus accounted for. Microcrystals permitting for serial data collection should have minimal dimensions, suitably sized to match the beam characteristics of the microfocus beamline. The dimensions of the crystals also define the minimal ligand diffusion time, which should be faster than the turnover time to reduce heterogeneity in the cryogenically trapped states

## Camera system

For automatic reference image acquisition and convenient sample alignment, the *spitrobot* is equipped with two cameras from

different viewing directions, 90° apart. The primary purpose of the camera system is sample alignment and LAMA-nozzle positioning. The axis of the LAMA nozzle is at an angle of 60° relative to the surface of the mesh. The LAMA nozzle can be positioned using the aforementioned 3-axis stage to optimally deposit the substrate on the micromesh and ensure reproducible results between different samples. By using the high-resolution camera system during sample preparation, the consistency between different samples can be maintained. In addition, this setup provides automatic image acquisition immediately before and after droplet depositions. This serves as a reference to (i) confirm the droplet depositions and (ii) droplet dissipation on the micromesh. The latter can be used during data collection at the beamline to narrow down the area for data collection (Supplementary Fig. 4).

### Serial crystallography

As a first *proof-of-principle* we focused on serial synchrotron data-collection (cryo-SSX). To demonstrate that spitrobot-prepared micro-crystals are suitable for cryo-SSX, 0.5 µl of the crystal slurry was directly loaded by pipette on the SPINE standard sample holder, 700/25 µm micromesh (MiTeGen, USA) and quickly transferred to the humidity stream in the *spitrobot* (Supplementary information). Excess mother liquor was manually blotted away until the sample meniscus disappeared, by quickly (<1 s) applying Whatman paper to the back of the mesh. For reaction initiation the ligand solution was supplied in the LAMA nozzle, 250–500 droplets were deposited in a burst mode (2–5 kHz repetition rate). For optimal comparison we used *Streptomyces rubiginosus* xylose isomerase (XI) (crystal size $10 \times 15 \times 15$ µm) and the *Klebsiella pneumoniae* extended spectrum β-lactamase CTX-M-14 (crystal size $20 \times 20 \times 20$ µm) as model systems, determining a ligand complex 50 ms and a covalent complex 1 s after reaction initiation, respectively (Fig. 2, Supplementary information). After a set delay time of 50 ms for xylose isomerase (XI) and 1 s for CTX-M-14 the crystals were vitrified in liquid nitrogen by directly plunging them into a puck. After structure determination clear difference electron density was visible in the active site, which could be interpreted by modeling the ligand molecules (Fig. 2).

Comparison to our previously determined CTX-M-14 structure in complex with avibactam (PDB-ID: 6GTH) reveals only minor differences between the cryogenically cooled crystals and the room-temperature complex determined by SFX[12]. This confirms efficient mixing and diffusion of the ligand into the active site as we have demonstrated previously[5,12] (Fig. 2, Supplementary Table 1).

Using 2,3-butanediol as a cryo-protectant in a buffer containing its natural ligand glucose, we found that 2,3-butanediol can also occupy the XI active site within 50 ms, and is not replaced by glucose within 500 ms after reaction initiation (Fig. 2, Supplementary Table 1). The 2,3-butanediol molecule soaked into the XI crystals adopts a conformation similar to our previously determined glucose bound complex structure 15 ms after reaction initiation[5].

### Canonical rotation crystallography

As an alternative to serial crystallography, we aimed to demonstrate that the *spitrobot* is also suitable for standard rotation data collection. We used the microfocus beam ($3 \times 7$ µm) of EMBL beamline P14 (PETRA III, Hamburg), where individual micro-crystals were centered in the X-ray beam. A convenient approach to identify well-diffracting crystals is generating a diffractive-power heat map via the mesh collection option in MXCuBE[13,14]. After selection of a suitable crystal a standard rotation dataset was collected, amenable to automatic data-processing routines available at most macro-molecular crystallography beamlines.

To demonstrate that such data collections work routinely, we prepared acyl-enzyme complexes of the activity impaired CTX-M-14 E166A mutant with ampicillin, at time-delays of 0.5 s, 1 s, and 5 s after reaction initiation. At all time points the electron density confirms that a covalent acyl-enzyme intermediate has formed, which compares well to previously published data and thus confirms the consistency of the crystallographic data across broad time-scales (7K2Y) (Fig. 2)[15]. In addition, this experiment demonstrates that cryo-trapping data can successfully be obtained via canonical rotation data collection and automatic data processing routines, which greatly accelerates the structure determination process.

Next, we aimed to explore the minimal permissible *spitrobot* delay time in comparison to previously established data[4]. For consistency between the results obtained via the LAMA method at room-temperature SSX and the cryo-trapping results from the *spitrobot* we made use of our previously established model system xylose isomerase (XI). XI microcrystals were loaded onto SPINE standard, 700/25 µm micromeshes (MiTeGen, USA), directly inside the humidity stream using a standard micro-pipette. For reaction initiation the substrate solution (1 M D-glucose (aq)) was supplied in the LAMA nozzle, 250 droplets were deposited using the burst mode (5 kHz repetition rate).

Previously we had determined by RT-SSX that near full ligand occupancy can be obtained in XI within 15 ms, exceeding what is mechanically feasible with the spitrobot[5]. Thus, crystals were vitrified after 50 ms, 250 ms, 500 ms, and 1000 ms to narrow down the practical vitrification time limits. Consistent with our previous results difference density for the glucose molecule could be observed in the XI active site consistently across all time-points. This emphasizes that fast delay times are accessible to the *spitrobot* and that biologically relevant time-scales in the millisecond time-domain can be addressed via cryo-trapping crystallography. This emphasizes the high reproducibility of the results, the comparability to TR-SSX data and the suitability for cryo-trapping experiments in the sub-second time-domain (Fig. 2).

### Cryo-trapping crystallography of tryptophan synthase reaction intermediates

Finally, we aimed to demonstrate as a *proof-of-principle* that the *spitrobot* can trap enzymatic reaction intermediates using single, macroscopic crystals. To this end, we used the *Salmonella typhimurium* tryptophan synthase (TS) (crystal size $200 \times 100 \times 50$ µm), which catalyzes the final steps in tryptophan biosynthesis. TS is a heterotetrameric pyridoxal 5′-phosphate (PLP) dependent bi-enzyme complex which generally assembles into a TrpA/TrpB$_2$/TrpA dimer of heterodimers. In this structural architecture, a ~25 Å long allosteric communication tunnel connects each of the TrpA/TrpB heterodimers[16,17]. The accepted model for the TS turnover reaction is as following: TrpA reversibly converts indole-3-glycerol phosphate (IGP) into glyceraldehyde-3-phosphate (G3P) and indole. Simultaneously, in the active site of TrpB, serine reacts with PLP forming an external aldimine (Aex-Ser) intermediate. Tryptophan is finally generated by indole passing through the tunnel between TrpA[18] and TrpB, allosterically regulated by the communication domain (COMM) of the TrpB subunit (Fig. 3a). Once indole reaches the TrpB active site it interacts with the Aex-Ser intermediate forming tryptophan as the end product, which is finally released from TS[17,19,20].

To demonstrate as a *proof-of-principle* that the *spitrobot* can be used to gain insight into reaction intermediates, TS macro-crystals were loaded onto standard 400/25 µm SPINE micromeshes (MiTeGen, USA) and mounted on the *spitrobot*. Turnover was initiated by LAMA-depositing reaction buffer containing indole, serine, and glyceraldehyde-3-phosphate (G3P) as substrates onto the TS crystals. To this end 500 droplets of reaction buffer were deposited at a 6 kHz repetition rate. The TS crystals were then vitrified in liquid nitrogen by

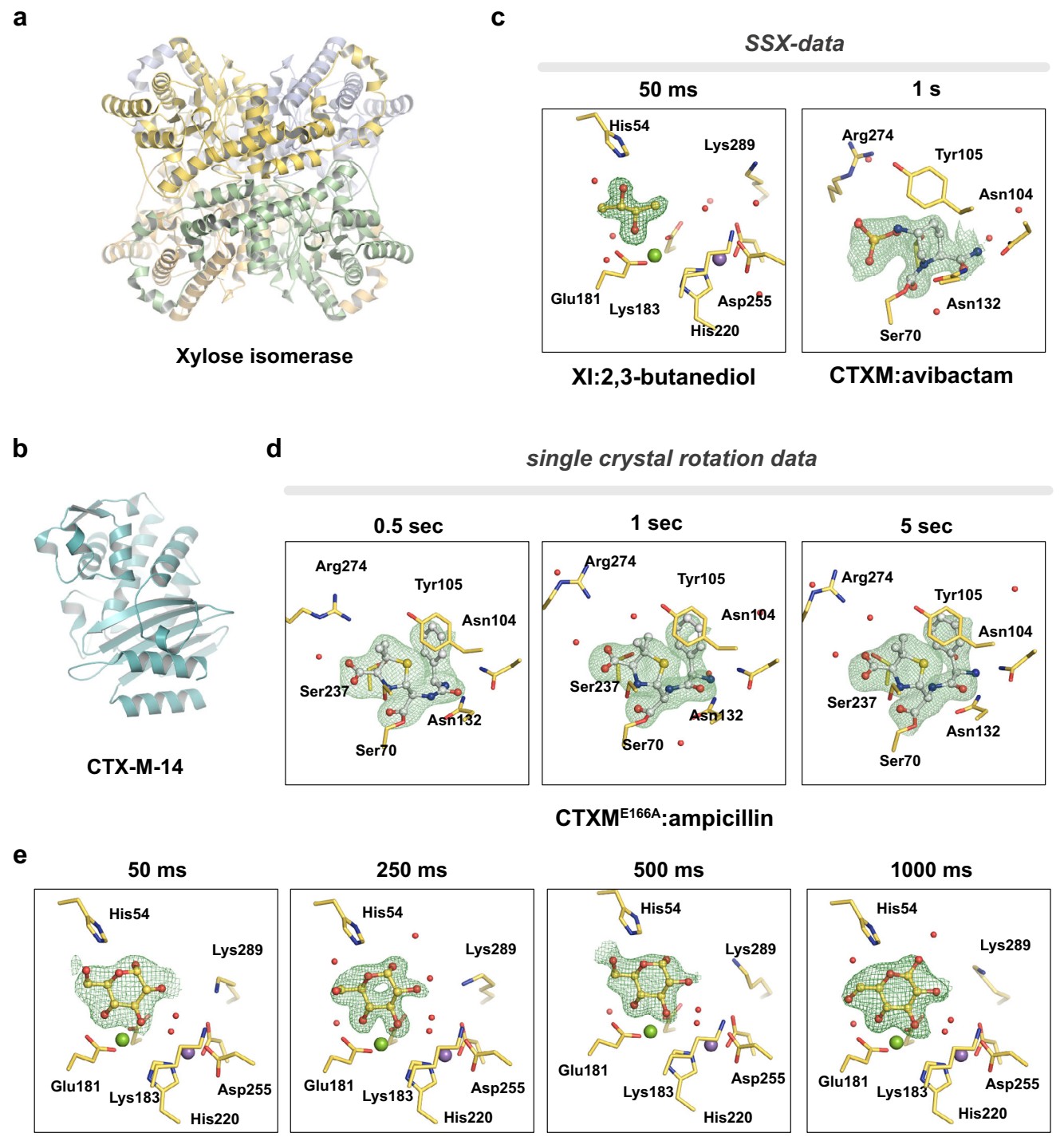

**Fig. 2 | Crystallographic assessment of representative time points.**
**a**, **b** Structural overview of Xylose isomerase (XI) and CTX-M-14 b-lactamase (**c**) cryo-SSX: XI 2,3-butanediol complex formation after 50 ms and CTX-M-14:avi-bactam complex formation after 1 s. **d**, **e** Single crystals: CTX-M-14$^{E166A}$:ampicillin complexes after 0.5, 1, and 5 s, respectively, and XI:glucose complexes after 50 ms, 250 ms, 500 ms, and 1000 ms after reaction initiation. POLDER omit maps are shown at 3.0 r.m.s.d.

plunging them directly into a puck at the specified delay time points (20 s, 25 s, and 30 s). At 20 and 30 s, in the active site of the TrpB subunit near full occupancy of the Aex-Ser intermediate are observed (Fig. 3). In addition to the formation of this intermediate, the side-chain movements of bLys87 and bGln114 are visible (Fig. 3b–e; Supplementary Fig. 8). The structure at 25 s visualizes the start of the β-subunit reaction. Here serine approaches the internal aldimine, priming the

β-subunit for the formation of the external aldimine (Aex-Ser), which can be clearly observed at 20 s and 30 s. Thus, the selected time-points presumably show snapshots of different cycles of the irreversible turnover reaction of the TS β-subunit. While further details will be addressed in more detail in a follow-up study, these results clearly demonstrate that the *spitrobot* can conveniently trap reaction intermediates in macroscopic crystals providing insight into enzymatic

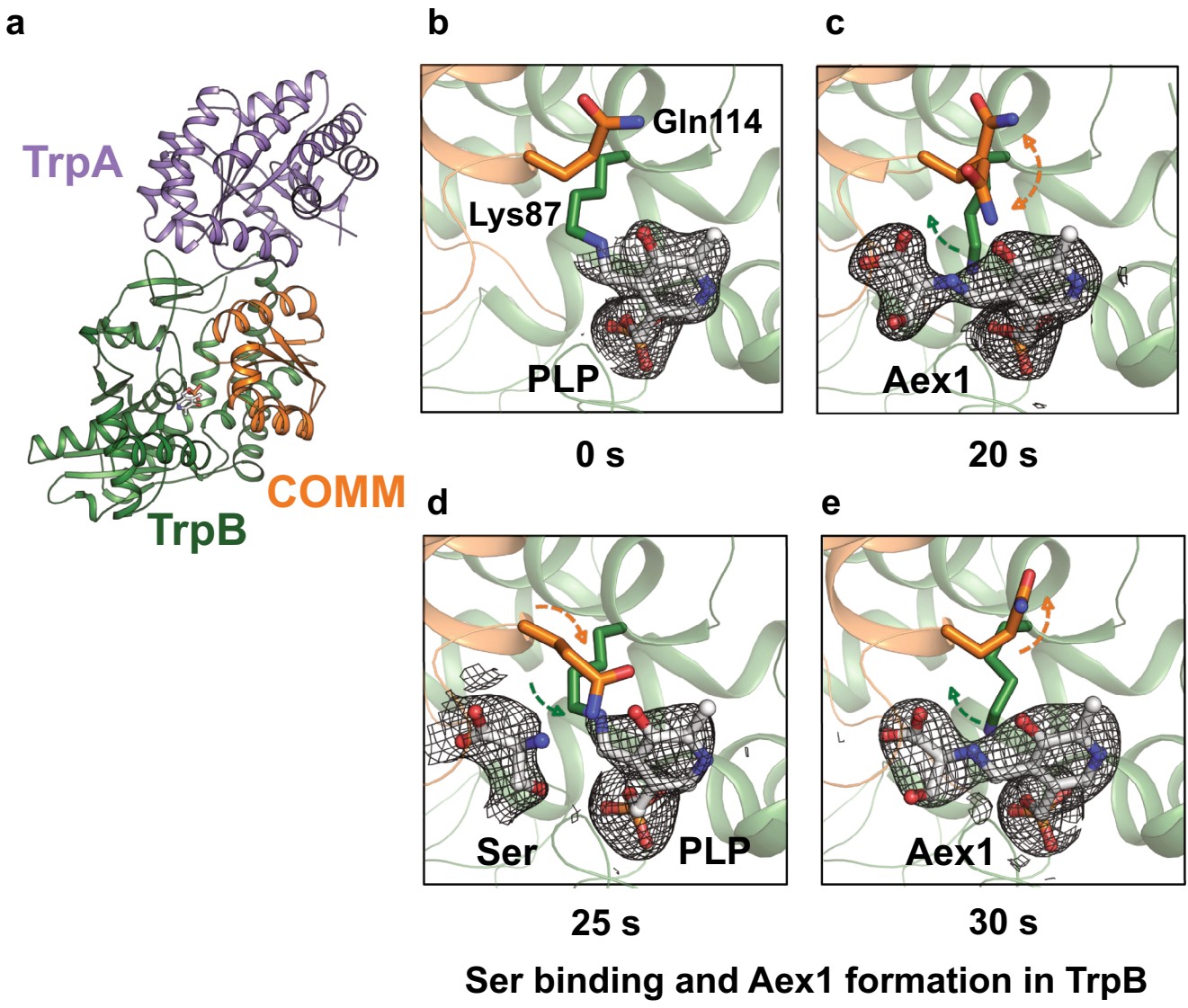

**Ser binding and Aex1 formation in TrpB**

**Fig. 3 | Time-resolved analysis of Tryptophan synthase (TS) turnover. a** Cartoon representation of Tryptophan synthase AB complex with each subunit represented in a different color. **b–e** Formation of an external aldimine intermediate (Aex1) at different time points (20 s, 30 s) and serine binding (25 s) after mixing. **c–e** rearrangement of the residues, Lys87 (green arrow) and Gln114 (orange arrow) during serine binding and Aex1 formation in TrpB, after reaction initiation. Active site residues of TrpB are represented as stick and substrates are represented as ball-and-stick. All electron density maps are represented as composite omit maps contoured at 1.0.

turnover. While the fast time-scales enabled by the spitrobot are inaccessible to manual procedures, the slow time-scales benefit from precision, accuracy and the overall experiment gains reproducibility.

## Discussion

Most available plungers are mainly intended to vitrify cryo-EM samples[21]. By contrast, the commercial plunger 'Nanuq' (Mitegen, USA) is specifically designed to vitrify protein crystals but can–*to the best of our knowledge*–not initiate reactions. More recently a similar crystal cryo-trapping solution was reported[10]. In contrast to the *spitrobot*, this 'mix-and-quench' device relies on non-standard crystal sample holders without bases and thus requires more complex crystal handling under cryo-conditions, which bears the potential for errors. Moreover, as the crystals are plunged through a substrate-containing film at high velocity it appears difficult to realize the biologically important long-time delays, which greatly limits its applicability to macroscopic crystals. By contrast, the *spitrobot* is built around the SPINE-standard, and versatile time delays have been demonstrated. While manual cryo-trapping is obviously an alternative approach for

larger crystals and longer time delays, reproducibility is a major issue with this approach, as time delays cannot be kept with high accuracy. Here, the relative error increases with decreasing time-delays and in consequence the comparability between different samples is reduced. While true time-resolved crystallographic experiments at room temperature offer a much more complete access to the dynamic landscape of protein function, cryo-trapping experiments may be sufficient to solve many important biologically relevant questions–e.g., provide insight into thermodynamically trapped, stable reaction intermediates[22,23]. It is clear that care has to be taken to only use crystals of highly similar dimensions to avoid that mechanistic conclusions are affected by different diffusion times. Along similar lines the utility of isomorphous difference maps could be affected by non-isomorphism originating in the freezing process.

However, if these aspects are accounted for *spitrobot* experiments provide a number of clear and important advantages. The *spitrobot* (i) allows for remote experiments by uncoupling of sample preparation from data-collection; (ii) makes sub-second cryo-trapping possible, which is manually either very hard or impossible, and produces longer

time delays more consistently than manual harvesting; (iii) enables preparation for RT time-resolved experiments with "a few" crystals to test, e.g., in situ mixing and investigate *in-crystal* kinetics; (iv) facilitates systems with an unfavorable crystal-size to diffraction ratio, unsuitable for serial crystallography; (v) provides physiological conditions during reaction initiation; (vi) permits low-brightness beamlines or home-sources to carry out experiments enabling insight into sub-second time-delays, as measuring cryo-trapped intermediates is less limited by the photon flux; (vii) provides lower susceptibility to radiation damage when collecting data from frozen crystals; (vii) offers controlled freezing, which may improve data-statistics; (ix) allows for cryo-SSX via collection of oscillation data, which is very sample efficient compared to RT serial approaches; (x) most importantly displays full compatibility with SPINE standards and thus with the existing high-throughput infrastructure available at most MX-beamlines.

While caution must be applied to mechanistic conclusions that could be biased by the cryo-trapping process (e.g., side-chain conformations, hydration structures, etc.) the practical advantages and its widespread applicability put the *spitrobot* into an ideal position for the transition from static structure determination to time-resolved crystallography projects at ambient temperatures[24]. The versatility of the spitrobot (crystal size, data-collection routines, time-delays, environmental control), provides ample target opportunities for a large number of labs and beamlines. Importantly the simplicity of the workflow including canonical data-processing, makes it also accessible to the inexperienced users.

## Methods

### Protein crystallization

**CTX-M-14**. CTX-M-14 crystals were generated as described previously[12]. Briefly: Purified CTX-M-14 was concentrated to 26 mg/ml and incubated with CTX-M crystallization buffer (40% PEG8000, 200 mM LiSO$_4$, 100 mM NaOAc, pH 4.5) and a highly concentrated seed stock in a 50:45:5 ratio for batch micro-crystallization of the protein. Homogenous micro-crystals with a typical size of $20 \times 20 \times 20\,\mu m$ were obtained within one day. The asymmetric unit contains one CTX-M-14 monomer in spacegroup P 3$_1$ 2 1. Crystals of the activity-impaired E166A mutant were generated under the same conditions.

**Xylose isomerase**. Macroscopic crystals of xylose isomerase (XI) were grown via the sitting drop vapor diffusion method. XI was concentrated to 40 mg/ml and equal volumes of protein solution and XI-crystallization buffer (31% (w/v) PEG 3350, 0.2 M LiSO$_4$, 0.01 M Hepes pH 7.5), were incubated for 4 days until first crystals formed, which were harvested after several weeks at a size of approximately 300 μm in diameter. The asymmetric unit contains one XI monomer in spacegroup I 2 2 2.

XI microcrystals were generated as described previously[5]. Briefly: Purified XI was concentrated to 80 mg/ml and incubated with equal amounts of XI crystallization buffer (35% (w/v) PEG3350, 0.2 M LiSO$_4$, 0.01 M Hepes pH 7.5). The solution was subjected to vacuum evaporation in a 'speedvac'-micro centrifuge (Eppendorf, Hamburg, Germany) for 15–20 min, yielding homogeneous micro-crystals with dimension of $10 \times 15 \times 15\,\mu m$.

**Tryptophan synthase (TS)**. TrpA and TrpB were purified as described previously[16]. Birefly: TrpAB was recombinantly expressed in *E. coli* (BL21 gold DE3) and purified via Ni-affinity chromatography (50 mM Tris/HCl pH 7.5, 150 mM NaCl, 10 mM imidazole; eluted with a linear imidazole gradient to 1 M) and an S75 size-exclusion column (50 mM Tris/HCl pH 7.5). The TS complex crystallized in 17% (w/v) PEG 300, 0.1 M Tris-HCl (pH 7.5), and 20 mM Cesium chloride. Crystals grew after mixing 2 μl at 8–9 mg/ml protein solution in size-exclusion buffer (50 mM Tris-HCl pH 7.5) with 2 μl of the reservoir solution at 18 °C by

the hanging-drop vapor diffusion method. Crystals appeared after 2–3 days and reached the final size ($200 \times 100 \times 50\,\mu m$) after five to seven additional days. The asymmetric unit contains one TrpA and one TrpB as TrpAB complex in spacegroup C 2.

### Reaction initiation

While microcrystal slurries (~500 nl) are deposited with a pipette, single-crystals are fished manually and quickly placed in the humidity stream on the spitrobot, excess mother liquor is manually blotted via Whatman paper. For reaction initiation the substrate solutions were sterile filtrated and degassed via sonication for 30 min. The substrate solutions were loaded into the LAMA nozzles according to the manufacturer's instructions (Microdrop Technologies, Norderstedt, Germany). For complex formation with CTX-M-14 500 150 pl droplets (~75 nl) of avibactam-buffer (0.5 M avibactam, 0.14 M LiSO$_4$, 0.07 M NaOAc, 0.006 M MES, 15% (v/v) 2,3-butanediol) or 500 droplets of ampicillin-buffer (1 M Na-ampicillin, 0.14 M LiSO$_4$, 0.07 M NaOAc, 0.006 M MES, 15% (v/v) 2,3-butanediol), respectively, were applied at a frequency of 2 kHz. For complex formation with XI 200-250 droplets of glucose-buffer (1 M Glucose) or 200–250 droplets of butanediol-buffer (1 M glucose, 15% (v/v) 2,3-butanediol), respectively, were applied at a frequency of 5 kHz. For reaction initiation with TS, 500 droplets of reaction-buffer (17% (w/v) PEG 300, 0.1 M Tris-HCl (pH 7.5), 20 mM Cesium chloride, 10 mM G3P, 110 mM indole, 100 mM serine, and 30% ethanol) were applied at a frequency of 6 kHz via the LAMA nozzle.

### Cryo protection

To avoid ice-crystal formation the substrate solutions were supplemented with an appropriate cryo-protectant. In the case of CTX-M-14 crystals 15% (v/v) 2,3-butanediol was combined with the substrate solution as stated above; in the case of XI crystals either a 15% (v/v) 2,3-butanediol solution or the 1 M glucose solution served as cryo-protectant; for TS the substrate solution contained 30% (v/v) ethanol.

### Data-collection and processing

**Cryo SSX**. Serial synchrotron crystallography was originally established under cryo-conditions using a limited rotation workflow[25]. However, unlike in the original workflow we collected still diffraction images using a mesh scan workflow available in MXCuBE[14]. A focused X-ray beam with a FWHM size of $7 \times 3\,\mu m$, at an energy of 12.7 keV (0.9763 Å) at a flux of ~$2 \times 10^{13}$ ph/s and an exposure time of 7.5 ms per image was used during data collection with an Eiger2 CdTe 16 M detector (Dectris, Switzerland). For the mesh collections, a mesh with a grid spacing matching the dimensions of the beam was drawn over the whole micro-mesh sample, giving rise to several thousand still diffraction images, which were processed using CrystFEL with the XGANDALF indexing routine[26,27]. Structures were solved by molecular replacement in PHASER using 6GTH as a search model for CTX-M-14 and 6RNF as a search model for XI[28].

**Single crystal data**. Cryo-trapping data from single crystals were solved by making use of canonical, single-crystal data-collection workflows. A focused beam with a FWHM size of $7 \times 3\,\mu m$, at an energy of 12.7 keV (0.9763 Å) at a flux of ~$4 \times 10^{11}$ ph/s and an exposure time of 7.5 ms per image was used during data-collection on an Eiger2 CdTe 16 M detector for P14 and the Eiger16M for P13 (Dectris, Switzerland). Diffraction data were processed using XDS[29–31] and AutoPROC using StarAniso[32,33]. For processing the TS datasets, the collected datasets were initially integrated using XDS and merged and scaled using the CCP4 suite program AIMLESS[34,35]. Structures were solved by molecular replacement in PHASER using 2WSY as a search model for TS[36], 6RNF as a search model for XI, and using 6GTH as a search model for CTX-M-14.

**Refinement and data analysis**. Refinement was carried out in the phenix suite using *phenix.refine*[37] and coot 0.8 for manual corrections to the model[38]. POLDER maps were generated using *phenix.polder*[39]. Composite omit maps for TS were generated using *phenix.composite_omit_map*. Molecular images were generated in PyMol[40].

### Reporting summary
Further information on research design is available in the Nature Portfolio Reporting Summary linked to this article.

## Data availability
The data that support this study are available from the corresponding authors upon request. All crystallographic data have been deposited in the Protein Data Bank (PDB) under accession codes 8AWE (Xylose Isomerase in 99% relative humidity), 8AWD (Xylose Isomerase in 95% relative humidity), 8AWB (Xylose Isomerase in 90% relative humidity), 8AWC (Xylose Isomerase in 85% relative humidity), 8AWF (Xylose Isomerase in 80% relative humidity), 8AW9 (Xylose Isomerase in 75% relative humidity), 8AW8 (Xylose Isomerase in 70% relative humidity). Data for the cryo SSX datasets of XI and CTXM-14, respectively, have been deposited under the accession numbers: 8AWY (Xylose Isomerase with 2,3-butanediol at 50 ms) and 8B3M (CTXM-14 Avibactam complex, SSX, 1 s). Data for the CTX-M-14[E166A] single crystal datasets have been deposited under the accession numbers: 8B2W (CTX-M-14 E166A, Ampicillin, 500 ms), 8B2V (CTX-M-14 E166A, Ampicillin, 1 s) and 8B2O (CTX-M-14 E166A, Ampicillin, 5 s). Data for the XI single crystal datasets have been deposited under the accession numbers: 8AWS (Xylose Isomerase with Glucose at 50 ms), 8AWU (Xylose Isomerase with Glucose at 250 ms), 8AWV (Xylose Isomerase with Glucose at 500 ms), 8AWX (Xylose Isomerase with Glucose at 1 s). Data for the TS single crystal datasets have been deposited under the accession numbers: 8B03 (Tryptophan synthase, 0 s), 8B05 (Tryptophan synthase, 20 s), 8B06 (Tryptophan synthase, 25 s), and 8B08 (Tryptophan synthase, 30 s). Previously published PDB files utilized can be found using accession codes 6GTH, 7K2Y, 2WSY, and 6RNF. Further details are available in Supplementary Tables 1–5.

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

## Acknowledgements

Data were collected at beamlines P13 and P14 operated by EMBL Hamburg at the PETRA III storage ring (DESY, Hamburg, Germany). We would like to thank our colleagues A.R. Pearson and G. Bourenkov for helpful discussions and critical reading of the manuscript. The authors gratefully acknowledge the support provided by the Max-Planck Society. P.M. acknowledges support from the Deutsche For-schungsgemeinschaft (DFG) via grant No. 451079909 and from a Joachim Herz Stiftung add-on fellowship. E.C.S. acknowledges sup-port by the DFG via grant No. 458246365, and by the Federal Ministry of Education and Research, Germany, under grant number 01KI2114. E.C.S., H.R., and A.P. acknowledge support from the Joachim Herz Stiftung via the Biomedical Physics of Infection Consortium. S.S. was supported by a fellowship from the EMBL Interdisciplinary Postdoc (EIPOD) program under Marie Sklodowska-Curie Actions COFUND (grant agreement number 664726).

## Author contributions

E.C.S. designed the experiment. E.C.S., P.M., and F.T. designed the spitrobot. E.C.S., P.M., and S.S. performed the experiments with support from D.v.S. P.M., A.P., A.B., and S.S. prepared the protein crystals. E.C.S., P.M., S.S., and D.v.S. processed and analyzed the diffraction data. C.E.H., S.K.-D., S.S., and G.G. refined crystal structures. J.-P.L., H.S., M.K., and F.T. built the spitrobot, the humidity flow device (HFD) and designed the LabView interfaces. M.W. and H.R. provided resources for the research and validated results. P.M. and E.C.S. wrote the manuscript. All authors discussed and corrected the manuscript.

## Funding

## Competing interests

On 10 March 2022, a patent application has been filed under the number EP22161384, for the Max-Planck Society and to F.T., E.C.S., H.S., P.M., M.K. under the following title: "VERFAHREN UND VORRICHTUNG ZUM BEREITSTELLEN VON BIOLOGISCHEN PROBEN IN EINEM VITRIFIZIERTEN ZUSTAND FÜR STATISCHE UND ZEITAUFGELÖSTE STRUKTUR-UNTERSUCHUNGEN MIT HILFE VON ELEKTRONEN- ODER RÖNTGEN-QUELLEN". The remaining authors declare no competing interests.
