## [Peer Review File · Nature Communications]

Millisecond cryo-trapping by the spitrobot crystal plunger simplifies time-resolved crystallography

Editorial Note: Parts of this Peer Review File have been redacted as indicated to remove third party material where no permission to publish were obtainedReviewers' Comments:

Reviewer #1:

Remarks to the Author:

The authors report on an interesting technological advancement in the field of serial protein X-ray crystallography. Specifically, they combine the previously applied LAMA ligand/substrate injector with a nitrogen plunger that provides extremely fast flash-cooling of either meshes with microcrystals or even of single crystals in the milliseconds-to-seconds time regime (depending on crystal size). As proof-of-concept, they employ their techniques to obtain snapshots of different enzymes in complex with intermediates evolving and accumulating during substrate binding and catalysis.

The method seems robust, and the extension of serial X-ray crystallography to single crystals will be of great use for the structural analysis of many enzymes in the future.

Even though the chemistry of the studied enzymes is well understood, it would help the readers if the bound substrates/intermediates were directly labeled in the figures.

Reviewer #2:

Remarks to the Author:

Millisecond cryo-trapping by the spitrobot crystal plunger simplifies time-resolved crystallography Mehrabi et al.

As protein crystallography is advancing from static protein structures to addressing essential dynamic aspects of protein function and capturing snapshots of proteins in action, methods and instrumentation that can be used by the large protein crystallography community for such studies are essential. This manuscript describes instrumentation that demonstrates possibility of cryo-trapping protein intermediates on ~50ms time scale when using diffusion-based reaction initiation. Described cryo-trapping is a trigger-freeze method to study protein intermediates that is certainly not new (Bourgeois, D., and Weik, M. (2009) Kinetic protein crystallography: a tool to watch proteins in action. *Crystallography Reviews* 15, 87–118). Although the method cannot examine all aspects of protein dynamics and cannot trap very fast intermediates, it is highly valuable for capturing slower intermediates, while using standard cryo data collection and standard data processing tools. Among a number of important advantages of the method described in the manuscript, a significant one is preliminary investigation of reaction initiation and protein kinetics in the crystal that can provide essential guidance to conducting more challenging time-resolved experiments at room temperature. Described method and achieved time resolution of 50ms are not completely novel as a similar trigger (by-diffusion)-freeze approach has been published recently (and is referenced in the manuscript; Clinger et al. (2021) *IUCrJ* 8, 784–792). However, I still recommend the manuscript for publication after addressing comments and questions below, mainly because it describes an integrated experimental setup that has a potential to be adopted and routinely used by a large crystallographic community interested in protein dynamics.

Comments, questions and suggestions:

Main text:

- It would be good to avoid using "time-resolved" when referring to the described cryo-trapping method. The authors largely do distinguish their method from the actual time-resolved method that they refer to as "true time-resolved". But they still unnecessarily use "time-resolved" at some places in the text in reference to their method.
- Freezing crystals typically requires adding cryoprotectant. Was it explored how cryoprotectant affects the substrate diffusion times? Were cryoprotectants used for the presented examples? Any comments on this topic could be useful.
- P. 2, lines 16, 21

Should include reference for the SPINE-standard.

- P. 2, lines 24-26

There should be a short summary here of what determines/limits the minimum achievable time delay of the instrument to ~50ms (with additional details provided in the Supplemental Notes).

- P.2, lines 27-36 – p.3, lines 1-4

Crystal size for samples used should be mentioned here. It is an important parameter as it affects both substrate diffusion time and vitrification delay/vitrification time, both related to the minimum time delay for trapping intermediates that can in principle be reached by this method.

- P.3, lines 10-11

It is not clear why 20-30s is “inaccessible to manual procedures.” Manual cryo-trapping should be possible on 5s time scale and certainly is possible on 10s and longer time scales.

- P. 3, lines 17-20

I suggest revising the following text that compares the spitrobot to another recently published ‘mix-and-quench’ device described in Clinger et al. (2021) IUCrJ 8, 784–792:

“Moreover, as the crystals are plunged through a substrate-containing film at high velocity it appears difficult to realise the biologically important long-time delays, which greatly limits its applicability to macroscopic crystals. By contrast, the spitrobot is built around the SPINE-standard, and versatile time-delays have been demonstrated.”

While it is true that spitrobot can easily and conveniently achieve long time delays, this may not be a particularly important advantage when comparing the two devices if Clinger et al method can achieve <5 sec time delays. The challenge for cryo-trapping are shortest, sub-second time delays. Very long delays of >5sec can be achieved even manually.

- P. 6, Reaction initiation

Are there other suitable LAMA nozzles that can deliver fewer but somewhat larger droplets rather than several hundreds of smaller droplets (which take tens or hundreds of ms to deliver)?

Figure captions:

- General – figure captions should be improved to provide sufficient information to understand the figures without the need to refer to the main text for some information shown in the figures. Example – figure 1d mentions 3 bar and 6 bar which is not addressed in the figure caption.

- Figure 1

Since the methods in the main text do not describe experimental characterization of the vitrification process, panel 1d should be moved to Supplemental Notes. Panel shows 250um and 13um sensor data while the figure caption mentions only a 15um sensor. Also, figure caption mentions 15um is a typical size of crystals while crystal sizes used for examples in the manuscript vary from micro-crystals of 10-15um to macro-crystals of 300um.

Supplemental Notes

Text

- P.3, line 1

HFD acronym should be defined before being used.

- P. 6, line 5

A short description of the Leidenfrost effect will be useful.

- P.6, line 8

It is not clear how “...a total time to quench of ~50 ms and is thus almost negligible for the relevant time scales” when relevant time scale is 50ms for some experiments.

- P.6, lines 12-16

When estimating 50ms as the minimum time scale for trapping intermediates by this instrument, I don't see that the measured and significant vitrification delay of 50ms (for small sample sizes of ~\$20um and significantly longer for larger samples) is taken into account. Only air valve delay of ~20 ms, piston motion of ~25 ms and actual vitrification process of ~7.5 ms are mentioned in defining ~50ms minimum time scale.

- P. 9, lines 27-28

Any possibility to automate the storage puck rotation?

- P.12, lines 22 and 25

Table 2 should be Table 1.

- Crystal sizes should be listed in Tables 2-5 or in corresponding parts of the text.

- P.19, line lines 5-6

Same as for P.3, lines 10-11 - it is not clear why 20-30s is "inaccessible via manual cryo-trapping approaches." Manual cryo-trapping should be possible on 5-10s time scale.

Reviewer #3:

Remarks to the Author:

The advent of X-ray free electron lasers (XFELs), next-generation synchrotrons has sparked new interest into following complex biological reactions using time-resolved structural biology. Typically, these experiments are performed at room temperature to allow conformational changes to happen in "real time". However, room temperature measurements also introduce serious experimental difficulties as radiation damage is more prominent and, even at XFELs, that can outrun most damage but are not suited for cryo-measurements, new sample has to be continuously replenished using the serial crystallography approach. The spitrobot presented in this manuscript provides an elegant solution as the reaction is first initiated at room temperature before it is quenched by freezing crystals after a preset time delay. The achievable time resolution in the millisecond range is sufficient to trap many interesting biological reactions and nicely complements XFEL experiments often targeting faster time domains. A wide adoption is ensured by making the device compatible with standard synchrotron setups and mounting of crystals on micromesh loops. The ability to collect oscillation series from frozen crystals should increase data quality and lower the number of crystals compared to other serial crystallography approaches. In the submitted manuscript the authors describe how reaction initiation can be achieved using rapid deposition of microliter-sized droplets similar to the solid support room temperature solution they had described before (Merhabi et al., Nature Methods, 2019). If the used crystals are small enough to allow for rapid diffusion, this approach fits well to the targeted millisecond time resolution. But even though it is not mentioned in the manuscript it should also be possible to use other triggers like optical lasers in combination with photosensitive proteins. The ability to control humidity, temperature as well as blotting time has further much potential to standardize freezing of crystals to improve reliability as well as diffraction properties as demonstrated using their Xylanase system. This could have many applications in conventional crystallography but is also important for time-resolved crystallography as the commonly used isomorphous difference maps are very sensitive to changes in crystal packing that can occur upon freezing. Overall, I congratulate the authors and would like to see the work published in a revised form as detailed below.

Where I struggle is the current ultra-short presentation of the work. The main text is only about 1000 words long and when I first read it, I thought nice introduction but where is the rest of the paper? I suggest to make better use of the space offered by Nature Communications and merge much of the supplementary material into the main manuscript. Here some concrete suggestions:

Abstract: Please name the enzymes you investigate and describe what can be seen. Millisecond time-resolution is somewhat misleading. Better use time-resolution in the millisecond range or subsecond time resolution. Currently the abstract is 68 out of allowed 150 words. It could convey more information on the studied systems and reactions.

Figure 2 is meant to demonstrate the subsecond time-resolution of the spitrobot. It would be important to show how the structures looked like before the reaction has been initiated, for example by using a control where buffer without substrate has been applied. Even though Figure 1 presents vitrification in the millisecond range, without the above control the manuscript does not really

experimentally demonstrate accessibility of the millisecond range. As presented all the time delays are basically the same. We need to see how it looks with and without ligand application to appreciate the changes and to be absolutely sure they originate from applying the ligand with the spitrobot. The experiments on the xylose isomerase nicely demonstrate the importance of controls as in case of the SSX data the cryo-protectant 2,3-butanediol was observed in the binding pocket and not the substrate glucose also in the buffer used to trigger the reaction. In contrast the datasets from the rotation of single XI crystals resolve the applied glucose.

In that respect, is there any difference in the shown time delays for the XI:glucose complex in Figure 2e? Have the authors tried occupancy refinement? If data was collected from a single or just a few crystals for each delay how would their size distribution effect diffusion rates? A 50 ms time delay collected using a 20 um sized crystal might look quite different compared to the same time delay collected using a 200 um sized crystal. How does the size contribution of crystals effect the collection of a whole time series? This should be much less of a problem when serial approaches are used, where each delay is collected from sometimes thousands of crystals. Please include some discussion. The consistency of electron density over several time delays as mentioned in the supplementary notes could be used as an argument if the authors have retained the information what size variation the used crystals had. A few words on crystal size are in the supplementary notes, somewhat out of context. Why split the paper in two if you are not short on space?

Currently it is very hard to understand much of the manuscript without digging through the supplementary notes. Much of the information on the systems and what is seen should be part of the main manuscript. How to appreciate a new method without seeing what can be done with it? For example, Figure 3 is meant to show the turnover within the tryptophane synthase. Arrows or similar indicators would help to appreciate structural rearrangements. What is the difference between c 20s and e 30s? The densities look very much the same with only 25s in between lacking a bond. It is not clear how this density shows the rearrangements for the reaction in Supplementary Figure 8. Another possibility would be to use Fobs-Fobs difference maps before and after substrate application. These should highlight rearrangements better compared to the used composite omit maps. The ability to trap enzymatic intermediates is an important application of the device and for me demonstrating this experimentally is an important reason to consider publication in Nature Communications instead of a more specialized, instrumentation focused journal. Please include a clearer description in the main manuscript so readers can follow to what extent the catalytic cycle has been resolved. Much of the material in the supplementary notes could be used for this.

In the list of advantages of the spitrobot where it is not always clear what has been compared. Point (v) physiological conditions during reaction initiation for example is unclear to me. How are conditions more physiological compared to time-resolved room temperature crystallography, how are they more physiological compared to trapping intermediates using cryoEM? Please clarify. Point (vi) reads ... as measuring cryo-trapped intermediates is not limited by the photon flux. Is that true? To see small changes upon trapping you want the best signal to noise ratio and resolution data. If you want high time resolution you need to use small crystals. Perhaps this should be just changed to less limited by photon flux.

As additional points the authors may also want to add to their list of advantages:

Lower susceptibility to radiation damage when collecting data from frozen crystals.

Controlled freezing may improve resolution as the authors show themselves in Supplementary Figure 6. In supplementary table 1 please add the relevant information to judge resolution i.e. Rmerge, I/sigma(I), CC 1/2 and completeness. At the moment it is not described how the resolution has been defined.

Could be easily used for pump-probe experiments using lasers as trigger system.

Collection of oscillation data is very sample efficient compared to serial crystallography.

In my understanding the collection of a oscillation series should provide superior quality data compared to datasets merged from single shots. However, this does not seem to be the case when comparing the data in supplementary table 2 and 4. Indeed there must be a mistake in Stable 4 where the 0.25s dataset has very high Rmerge values. Also the Rmerge for the 0.05s dataset is very high but probably still acceptable. What is the resolution criterion used here? If one wants to follow a time series the data should be treated as similar as possible. Number of reflections and number of unique reflections is missing for the 0.25s dataset. Rwork / Rfree for the 0.025s and 0.05s datasets have mistakes. Please add the number of crystals used to all oscillation datasets. The size of the used crystals would also be valuable to judge the diffusion time considerations mentioned above.

Some disadvantages would be:

Non-isomorphous unit cells after freezing prevents the use of Fobs-Fobs difference maps.

The effect of crystal size on time resolution and the information content in time series as mentioned above.

At the position in the main text where the authors describe how sample images are automatically aquired the authors may want to refer to Supplementary Figure 4. i.e

"As a quality control, sample images are automatically acquired before and after droplet deposition (Supplementary Figure 4)."

In Supplementary Figure 4 it would be great to include size bars and arrows highlighting individual crystals and perhaps the applied droplet.

Millisecond cryo-trapping by the spitrobot crystal plunger simplifies time-resolved crystallography

REVIEWER COMMENTS

Response to all reviewers:

While reviewer remarks are in *black sans-serif font italics*, our answers are in *blue serif-font*.
As requested by reviewer #3 several sections of the supplemental material have been merged with the main text in the revised version of the manuscript. The individual questions have been numbered chronologically.

Reviewer #1 (Remarks to the Author):

The authors report on an interesting technological advancement in the field of serial protein X-ray crystallography. Specifically, they combine the previously applied LAMA ligand/substrate injector with a nitrogen plunger that provides extremely fast flash-cooling of either meshes with microcrystals or even of single crystals in the milliseconds-to-seconds time regime (depending on crystal size). As proof-of-concept, they employ their techniques to obtain snapshots of different enzymes in complex with intermediates evolving and accumulating during substrate binding and catalysis.

The method seems robust, and the extension of serial X-ray crystallography to single crystals will be of great use for the structural analysis of many enzymes in the future.

1. *Even though the chemistry of the studied enzymes is well understood, it would help the readers if the bound substrates/intermediates were directly labeled in the figures.*

#1 We appreciate the reviewers' comments and have updated the figures accordingly.

Reviewer #2 (Remarks to the Author):

Millisecond cryo-trapping by the spitrobot crystal plunger simplifies time-resolved crystallography

As protein crystallography is advancing from static protein structures to addressing essential dynamic aspects of protein function and capturing snapshots of proteins in action, methods and instrumentation that can be used by the large protein crystallography community for such studies are essential. This manuscript describes instrumentation that demonstrates possibility of cryo-trapping protein intermediates on ~50ms time scale when using diffusion-based reaction initiation. Described cryo-trapping is a trigger-freeze method to study protein intermediates that is certainly not new (Bourgeois, D., and Weik, M. (2009) Kinetic protein crystallography: a tool to watch proteins in action. Crystallography Reviews 15, 87–118). Although the method cannot examine all aspects of protein dynamics and cannot trap very fast intermediates, it is highly valuable for capturing slower intermediates, while using standard cryo data collection and standard data processing tools. Among a number of important advantages of the method described in the manuscript, a significant one is preliminary investigation of reaction initiation and protein kinetics in the crystal that can provide essential guidance to conducting more challenging time-resolved experiments at room temperature. Described method and achieved time resolution of 50ms are not completely novel as a similar trigger (by-diffusion)-freeze approach has been published recently (and is referenced in the manuscript; Clinger et al. (2021) IUCrJ 8, 784–792). However, I still recommend the manuscript for publication after addressing comments and questions below, mainly because it describes an integrated experimental setup that has a potential to be adopted and routinely used by a large crystallographic community interested in protein dynamics.

We appreciate the reviewers comment and have added a citation to Bourgeois, D., and Weik, M. (2009).

Comments, questions and suggestions:

Main text:

- 2. It would be good to avoid using “time-resolved” when referring to the described cryo-trapping method. The authors largely do distinguish their method from the actual time-resolved method that they refer to as “true time-resolved”. But they still unnecessarily use “time-resolved” at some places in the text in reference to their method.*

Corrected as suggested.

- 3. Freezing crystals typically requires adding cryoprotectant. Was it explored how cryoprotectant affects the substrate diffusion times? Were cryoprotectants used for the presented examples? Any comments on this topic could be useful.*

We appreciate that the reviewer raises this important topic. Although mentioned in the material and methods section we have added a dedicated section to the manuscript:

“Cryo protection

To avoid ice-crystal formation the substrate solutions were supplemented with an appropriate cryo-protectant. In the case of CTX-M-14 crystals 15% (v/v) 2,3-butanediol was combined with the substrate solution as stated above; in the case of XI crystals either a 15% (v/v) 2,3-butanediol solution or the 1M glucose solution served as cryoprotectant; for TS the substrate solution contained 30% (v/v) ethanol.”

4. P. 2, lines 16, 21
Should include reference for the SPINE-standard.

Added

5. P. 2, lines 24-26
There should be a short summary here of what determines/limits the minimum achievable time delay of the instrument to ~50ms (with additional details provided in the Supplemental Notes).

Following sentence has been added to the main text:

„The minimal delay time currently achievable with the spitrobot setup is ~50 ms. This constraint is the result of the mechanical piston motion (22.5 ms), air-valve delay (20 ms), and vitrification time (7.5 ms) – the deposition of the substrate solution occurs during the air-valve delay and is thus accounted for.”

6. P.2, lines 27-36 – p.3, lines 1-4
Crystal size for samples used should be mentioned here. It is an important parameter as it affects both substrate diffusion time and vitrification delay/vitrification time, both related to the minimum time delay for trapping intermediates that can in principle be reached by this method.

We have added the relevant information into the main text.

7. P.3, lines 10-11
It is not clear why 20-30s is “inaccessible to manual procedures.” Manual cryo-trapping should be possible on 5s time scale and certainly is possible on 10s and longer time scales.

Manual cryo-trapping is indeed an alternative approach for larger crystals and longer time-delays. However, reproducibility is a major issue with this approach as time-delays can never be kept with high accuracy. The relative error becomes larger the shorter the time-delay. In consequence the comparability between different samples – even at the same delay time – is reduced.

The discussion has been updated accordingly:

“While manual cryo-trapping is obviously an alternative approach for larger crystals and longer time delays, reproducibility is a major issue with this approach, as time delays cannot be kept with high accuracy. Here, the relative error increases with decreasing time-delays and in consequence the comparability between different samples is reduced.”

8. P. 3, lines 17-20
*I suggest revising the following text that compares the spitrobot to another recently published ‘mix-and-querch’ device described in Clinger et al. (2021) IUCrJ 8, 784–792:
“Moreover, as the crystals are plunged through a substrate-containing film at high velocity it appears difficult to realise the biologically important long-time delays, which greatly limits its applicability to macroscopic crystals. By contrast, the spitrobot is built around the SPINE-standard, and versatile time-delays have been demonstrated.”
While it is true that spitrobot can easily and conveniently achieve long time delays, this may not be a particularly important advantage when comparing the two devices if Clinger et al method can achieve <5 sec time delays. The challenge for cryo-trapping are shortest, sub-second time delays. Very long delays of >5sec can be achieved even manually.*

Please see our comment above (#7) regarding manual trapping.

9. *P. 6, Reaction initiation*

Are there other suitable LAMA nozzles that can deliver fewer but somewhat larger droplets rather than several hundreds of smaller droplets (which take tens or hundreds of ms to deliver)?

The LAMA nozzles are commercially available in 3 different sizes, 30 μm , 50 μm , and 70 μm . The largest nozzles produces droplets of approximately twice the volume used within this manuscript and also provide better compatibility with more viscous solutions, which can be helpful for e.g. cryo-protectant solutions. However, the number of droplets that are required are also affected by the droplet distribution on the mesh-loops, which is generally affected by the viscosity of the solution and its surface tension.

Figure captions:

10. *General – figure captions should be improved to provide sufficient information to understand the figures without the need to refer to the main text for some information shown in the figures. Example – figure 1d mentions 3 bar and 6 bar which is not addressed in the figure caption.*

The figure caption has been updated.

11. *Figure 1*

Since the methods in the main text do not describe experimental characterization of the vitrification process, panel 1d should be moved to Supplemental Notes. Panel shows 250um and 13um sensor data while the figure caption mentions only a 15um sensor. Also, figure caption mentions 15um is a typical size of crystals while crystal sizes used for examples in the manuscript vary from micro-crystals of 10-15um to macro-crystals of 300um.

We fixed the figure caption and have updated the main text extensively, to include the vitrification process. Therefore, we feel that including figure 1d in the main text is appropriate.

Supplemental Notes

12. *P.3, line 1*

HFD acronym should be defined before being used.

Defined:

„The *spitrobot* comprises several different, main hardware parts: a) the plunger, b) the humidity flow device (HFD), c) the LAMA droplet injector, d) the vitrification chamber, f) the camera system, and e) the control unit. All parameters are set via a control software.“

„To maintain the crystals in an environment close to physiological conditions with controlled humidity and temperature we have developed a Humidity Flow Device (HFD), which provides a humid airflow at a defined temperature between 4 °C and 40 °C (**Supplementary Figure 2**).“

13. *P. 6, line 5*

A short description of the Leidenfrost effect will be useful.

Added:

„[...] which is presumably due to the Leidenfrost effect (insulating vapor layer between the sample and the cryogenic liquid), as the temperature drop-off from quenching is quite quick once the insulating vapor layer has dissipated.“

14. P.6, line 8

It is not clear how “...a total time to quench of ~50 ms and is thus almost negligible for the relevant time scales” when relevant time scale is 50ms for some experiments.

To avoid further confusion, we have added the term of ‘processing time’ to the SI:

„Taking into account the spitrobots total processing time, including air-valve delay, plunge time, and vitrification, the larger RTD (3.0 x 0.8 x 0.25 mm (IST, P1K0.308.7W.B.007 Farnell, Germany, -200°C – 600°C) displays a total time of 800 ms, due to a large offset which is presumably due to the Leidenfrost effect (insulating vapor layer between the sample and the cryogenic liquid), as the temperature drop-off from quenching is quite quick once the insulating vapor layer has dissipated. By contrast, the smaller thermocouple, which has approximately the same dimensions as the samples of interest (K-type thermocouple [KFT-13-200-200(Y)], ~13 µm diameter, ANBE SMT Co., Osaka, Japan) minimizes this offset and therefore displays a total processing time of ~50 ms and is thus almost negligible for the relevant time scales (Figure 1d). To determine the actual vitrification time, we also recorded the temperature decrease independent of the spitrobot. Here the glass-transition temperature (< -140 °C) is reached within 7.5 ms and a 90-10% analysis of the temperature drop resulted in a fall time of 7.5 ms. Thus, the total delay time of the spitrobot consists of the intrinsic delay of the device (air valve delay ~20 ms, piston motion ~25 ms), and the actual vitrification process ~7.5 ms corresponding to a cooling rate of $2.3 \times 10^4 \text{ Ks}^{-1}$. These vitrification times are comparable to those reported previously for flash cooling devices operating with liquid nitrogen¹. Based on these observations the dead-time of the *spitrobot* is on the order of ~45 ms and time-points with a minimal delay time of approximately 50 ms can be obtained.“

15. P.6, lines 12-16

When estimating 50ms as the minimum time scale for trapping intermediates by this instrument, I don't see that the measured and significant vitrification delay of 50ms (for small sample sizes of ~20µm and significantly longer for larger samples) is taken into account. Only air valve delay of ~20 ms, piston motion of ~25 ms and actual vitrification process of ~7.5 ms are mentioned in defining ~50ms minimum time scale.

The relevant aspect for trapping reaction intermediates is the vitrification time. During this period, the sample cools down to the glass-transition temperature when all molecular motions come to rest. This period is 7.5 ms. During this period thermodynamic trapping of biochemical reactions occurs. That is in contrast to RT time-resolved crystallography (SSX, SFX), cryo-trapping with the spitrobot does not allow to resolve meta-stable, that is kinetically trapped intermediates.

To emphasize the difference between vitrification time and the total required sample-preparation time, we have introduced the ‘total processing time’. This latter period includes all steps, that is air-valve delay, plunge time, and vitrification.

16. P. 9, lines 27-28

Any possibility to automate the storage puck rotation?

In the current setup this appears to be a rather complicated to achieve mechanically, since moving parts under cryogenic conditions can lead to seizing.

17. P.12, lines 22 and 25

Table 2 should be Table 1.

Crystal sizes should be listed in Tables 2-5 or in corresponding parts of the text.

Crystal sizes were added to the text.

18. P.19, line lines 5-6

Same as for P.3, lines 10-11 - it is not clear why 20-30s is "inaccessible via manual cryo-trapping approaches." Manual cryo-trapping should be possible on 5-10s time scale.

Please refer to out answer to question nr. 7.

Reviewer #3 (Remarks to the Author):

The advent of X-ray free electron lasers (XFELs), next-generation synchrotrons has sparked new interest into following complex biological reactions using time-resolved structural biology. Typically, these experiments are performed at room temperature to allow conformational changes to happen in “real time”. However, room temperature measurements also introduce serious experimental difficulties as radiation damage is more prominent and, even at XFELs, that can outrun most damage but are not suited for cryo-measurements, new sample has to be continuously replenished using the serial crystallography approach. The spitrobot presented in this manuscript provides an elegant solution as the reaction is first initiated at room temperature before it is quenched by freezing crystals after a preset time delay. The achievable time resolution in the millisecond range is sufficient to trap many interesting biological reactions and nicely complements XFEL experiments often targeting faster time domains. A wide adoption is ensured by making the device compatible with standard synchrotron setups and mounting of crystals on micromesh loops. The ability to collect oscillation series from frozen crystals should increase data quality and lower the number of crystals compared to other serial crystallography approaches. In the submitted manuscript the authors describe how reaction initiation can be achieved using rapid deposition of microliter-sized droplets similar to the solid support room temperature solution they had described before (Merhabi et al., Nature Methods, 2019). If the used crystals are small enough to allow for rapid diffusion, this approach fits well to the targeted millisecond time resolution. But even though it is not mentioned in the manuscript it should also be possible to use other triggers like optical lasers in combination with photosensitive proteins. The ability to control humidity, temperature as well as blotting time has further much potential to standardize freezing of crystals to improve reliability as well as diffraction properties as demonstrated using their Xylanase system. This could have many applications in conventional crystallography but is also important for time-resolved crystallography as the commonly used isomorphous difference maps are very sensitive to changes in crystal packing that can occur upon freezing. Overall, I congratulate the authors and would like to see the work published in a revised form as detailed below.

Where I struggle is the current ultra-short presentation of the work. The main text is only about 1000 words long and when I first read it, I thought nice introduction but where is the rest of the paper? I suggest to make better use of the space offered by Nature Communications and merge much of the supplementary material into the main manuscript. Here some concrete suggestions:

- 19. Abstract: Please name the enzymes you investigate and describe what can be seen. Millisecond time-resolution is somewhat misleading. Better use time-resolution in the millisecond range or subsecond time resolution. Currently the abstract is 68 out of allowed 150 words. It could convey more information on the studied systems and reactions.*

Corrected as suggested.

- 20. Figure 2 is meant to demonstrate the subsecond time-resolution of the spitrobot. It would be important to show how the structures looked like before the reaction has been initiated, for example by using a control where buffer without substrate has been applied. Even though Figure 1 presents vitrification in the millisecond range, without the above control the manuscript does not really experimentally demonstrate accessibility of the millisecond range. As presented all the time delays are basically the same. We need to see how it looks with and without ligand application to appreciate the changes and to be absolutely sure they originate from applying the ligand with the spitrobot. The experiments on the xylose isomerase nicely demonstrate the importance of controls as in case of the SSX data the cryo-protectant 2,3-butanediol was observed in the binding pocket and not the substrate glucose also in the buffer used to trigger the reaction. In contrast the datasets from the rotation of single XI crystals resolve the applied glucose.*

The buffer control the reviewer asks for is shown for Tryptophane Synthase (Figure 3). Here the 0s intermediate does not contain the serine, which is clearly visible e.g. in the 25s time-point and as part of the Aex-1 intermediate visible at 20s and 30s demonstrating that buffer components diffuse into the active site. There are numerous apo (ligand-free) structures of xylose isomerase and CTX-M-14 available in the PDB. Since there are no major structural changes occurring upon ligand binding, we have refrained from showing this in the manuscript. Moreover, we have recently probed the diffusion time and occupancy refinement using the same xylose isomerase crystals in the original LAMA paper (Mehrabi et al 2019, Nat. Methods), where we have demonstrated that near full occupancy can be obtained close to theoretical diffusion times (15 ms). As the spitrobot is ‘mechanically limited’ to time-delays of about 50 ms, we chose this as our first time-delay point.

To emphasize the equivalence between the apo and ligand-bound states we would like to refer the reviewer to the supplemental information of *Mehrabi et al 2019, Nat. Methods*, which we are showing below:

REDACTED

Editorial Note: Parts of this Peer Review File have been redacted as indicated to remove third party material where no permission to publish were obtained

Superposition of XI structures over time – taken from Mehrabi et al 2019, Nat. Methods

- a) Cartoon representation of the 7 XI-monomer structures superimposed with an overall r.m.s.d of 0.09 \AA^2 indicating that differences between all the XI structures are within error.
- b) The active site of all the superimposed XI structures showing that the side chains as well as the structural waters within a 7 \AA radius around the ligand also show near identical conformations from one time point to the next.

21. *In that respect, is there any difference in the shown time delays for the XI:glucose complex in Figure 2e? Have the authors tried occupancy refinement?*

We show four different structures of a time-course of 1000 ms to demonstrate the consistency in the binding mode and the reproducible quality of the electron density in four independent experiments. Xylose isomerase is a hyperthermophile enzyme with practically no activity at room temperature. The binding mode at this temperature is constant over several hours (*unpublished*) and only by an increase in temperature catalytic activity can be induced (Mehrabi et al. bioRxiv 2021 <https://doi.org/10.1101/2021.11.07.467596>). Indeed there are no major structural changes visible in the XI:glucose complex structures over the presented time-course. Minor conformational changes in the glucose molecules are visible, however – these can rather be associated to variance in the quality of the electron density.

22. *If data was collected from a single or just a few crystals for each delay how would their size distribution effect diffusion rates? A 50 ms time delay collected using a 20 um sized crystal might look quite different compared to the same time delay collected using a 200 um sized crystal. How does the size contribution of crystals effect the collection of a whole time series?*

Data for serial data was collected from several thousand highly homogenous crystals. Single crystal rotation data were collected only from a single crystal. While the shortest crystal dimension is relevant with respect to the diffusion time, the reviewer is correct to assume that crystal size will affect the diffusion time, in particular if the size difference is on an order of magnitude as the reviewer suggests. Therefore, it is mandatory to only use crystals, which are highly homogenous in their size to avoid comparing samples with different diffusion times. To emphasize this aspect, we have added following sentence to the manuscript:

“It is clear that care has to be taken to only use crystals of highly similar dimensions to avoid that mechanistic conclusions are affected by different diffusion times.”

23. *This should be much less of a problem when serial approaches are used, where each delay is collected from sometimes thousands of crystals. Please include some discussion. The consistency of electron density over several time delays as mentioned in the supplementary notes could be used as an argument if the authors have retained the information what size variation the used crystals had. A few words on crystal size are in the supplementary notes, somewhat out of context. Why split the paper in two if you are not short on space? Currently it is very hard to understand much of the manuscript without digging through the supplementary notes. Much of the information on the systems and what is seen should be part of the main manuscript. How to appreciate a new method without seeing what can be done with it?*

The paper was originally prepared for a different Nature Journal and passed on to Nature Communications by direct transfer. We appreciate the reviewers desire for a more detailed description and have moved various sections of the SI into the main text. The crystal size is reported in the material and methods section. The homogeneity of the crystals as determined via light-microscopy is indeed very high.

24. *For example, Figure 3 is meant to show the turnover within the tryptophane synthase. Arrows or similar indicators would help to appreciate structural rearrangements. What is the difference between c 20s and e 30s?*

We have labelled the structural intermediates in Fig.3 and added arrows to emphasize structural rearrangements in Supplementary Fig. 8. Indeed, Fig 3c (20s) and Fig 3e (30s) are almost identical. Our current hypothesis is that we observe several catalytic cycles in the TS reaction mechanism, which agrees well with kinetic data. This is highly similar to a previous study, where we observed several catalytic cycles in the turnover of an enzyme (Mehrabi et al 2019, Science)

25. *The densities look very much the same with only 25s in between lacking a bond. It is not clear how this density shows the rearrangements for the reaction in Supplementary Figure 8. Another possibility would be to use Fobs-Fobs difference maps before and after substrate application. These should highlight rearrangements better compared to the used composite omit maps. The ability to trap enzymatic intermediates is an important application of the device and for me demonstrating this experimentally is an important reason to consider publication in Nature Communications instead of a more specialized, instrumentation focused journal. Please include a clearer description in the main manuscript so readers can follow to what extent the catalytic cycle has been resolved. Much of the material in the supplementary notes could be used for this.*

We appreciate the reviewer's interest and have merged the TS passages from the SI into the main text to give the readers a clearer view of the catalytic cycle. In addition, to avoid any ambiguity with respect to structural changes of the side chains and to make the figure in the main text clearer, we have merged Suppl. Fig8 with Fig3 in the main text, highlighting the side-chain rearrangements, which have been trapped via the spitrobot – as suggested by the reviewer.

26. *In the list of advantages of the spitrobot where it is not always clear what has been compared. Point (v) physiological conditions during reaction initiation for example is unclear to me. How are conditions more physiological compared to time-resolved room temperature crystallography, how are they more physiological compared to trapping intermediates using cryoEM? Please clarify.*

The humidity control system not only enables a modification of the environmental humidity but also setting of the temperature within a range of ~5°C to ~40°C. Most time-resolved experiments are simply done at RT, although usually without a clear definition of the actual temperature, this often corresponds to 20°C – 25°C. However, many interesting and/or medically relevant enzymatic reactions take place at e.g. human body temperature. Setting the spitrobot to 37°C the samples can be kept at this physiological temperature and thereby provide insight into conditions that are closer to the enzyme's physiological environment, before they are vitrified.

27. *Point (vi) reads ... as measuring cryo-trapped intermediates is not limited by the photon flux. Is that true? To see small changes upon trapping you want the best signal to noise ratio and resolution data. If you want high time resolution you need to use small crystals. Perhaps this should be just changed to less limited by photon flux.*

Changed accordingly.

28. *As additional points the authors may also want to add to their list of advantages: Lower susceptibility to radiation damage when collecting data from frozen crystals. Controlled freezing may improve resolution as the authors show themselves in Supplementary Figure 6. Collection of oscillation data is very sample efficient compared to serial crystallography.*

We have updated the list of advantages:

“However, if these aspects are accounted for *spitrobot* experiments provide a number of clear and important advantages. The *spitrobot* (i) allows for remote experiments by uncoupling of sample preparation from data-collection; (ii) makes sub-second cryo-trapping possible, which is manually either very hard or impossible, and produces longer time delays more consistently than manual harvesting; (iii) enables preparation for RT time-resolved experiments with “a few” crystals to test, e.g., *in-situ* mixing and investigate *in-crystal* kinetics; (iv) facilitates systems with an unfavorable crystal-size to diffraction ratio, unsuitable for serial crystallography; (v) provides physiological conditions during reaction initiation; (vi) permits low-brightness beamlines or home-sources to carry out

experiments enabling insight into sub-second time-delays, as measuring cryo-trapped intermediates is less limited by the photon flux; (vii) provides lower susceptibility to radiation damage when collecting data from frozen crystals; (viii) offers controlled freezing, which may improve data-statistics; (ix) allows for cryo-SSX via collection of oscillation data, which is very sample efficient compared to RT serial approaches; (x) most importantly displays full compatibility with SPINE standards and thus with the existing high-throughput infrastructure available at most MX-beamlines. ”

29. *In supplementary table 1 please add the relevant information to judge resolution i.e. Rmerge, I/sigma(I), CC 1/2 and completeness. At the moment it is not described how the resolution has been defined.*

We have relied on a 30% CC1/2 cutoff criterion. Supp. Table1 has been updated according to the reviewer’s suggestion.

30. *Could be easily used for pump-probe experiments using lasers as trigger system.*

This is work currently in progress and will be part of a separate publication.

31. *In my understanding the collection of a oscillation series should provide superior quality data compared to datasets merged from single shots. However, this does not seem to be the case when comparing the data in supplementary table 2 and 4. Indeed there must be a mistake in Stable 4 where the 0.25s dataset has very high Rmerge values. Also the Rmerge for the 0.05s dataset is very high but probably still acceptable. What is the resolution criterion used here? If one wants to follow a time series the data should be treated as similar as possible. Number of reflections and number of unique reflections is missing for the 0.25s dataset. Rwork / Rfree for the 0.025s and 0.05s datasets have mistakes. Please add the number of crystals used to all oscillation datasets. The size of the used crystals would also be valuable to judge the diffusion time considerations mentioned above.*

We are grateful to the reviewer’s careful inspection and have updated the tables accordingly. Unless otherwise stated we only used a single crystal for the oscillation datasets. The crystal sizes used for the various experiments is given in the materials and methods section.

32. *Some disadvantages would be: Non-isomorphous unit cells after freezing prevents the use of Fobs-Fobs difference maps. The effect of crystal size on time resolution and the information content in time series as mentioned above.*

The main text of the manuscript has been updated with following section:

”It is clear that care has to be taken to only use crystals of highly similar dimensions to avoid that mechanistic conclusions are affected by different diffusion times. Along similar lines the utility of isomorphous difference maps could be affected by non-isomorphism originating in the freezing process.”

33. *At the position in the main text where the authors describe how sample images are automatically aquired the authors may want to refer to Supplementary Figure 4. i.e. “As a quality control, sample images are automatically acquired before and after droplet deposition (Supplementary Figure 4).” In Supplementary Figure 4 it would be great to include size bars and arrows highlighting individual crystals and perhaps the applied droplet.*

Updated as suggested.